# Shoulder Positioning during Superior Capsular Reconstruction: Computational Analysis of Graft Integrity and Shoulder Stability

**DOI:** 10.3390/biology10121263

**Published:** 2021-12-03

**Authors:** Madalena Antunes, Carlos Quental, João Folgado, Clara de Campos Azevedo, Ana Catarina Ângelo

**Affiliations:** 1IDMEC, Instituto Superior Técnico, Universidade de Lisboa, Av, Rovisco Pais, 1049-001 Lisboa, Portugal; madalena.antunes@tecnico.ulisboa.pt (M.A.); jfolgado@tecnico.ulisboa.pt (J.F.); 2Life and Health Sciences Research Institute (ICVS), School of Medicine, University of Minho, Campus de Gualtar, 4710-057 Braga, Portugal; claracamposazevedo@gmail.com; 3ICVS/3B’s-PT Government Associate Laboratory, 4805-017 Braga/Guimarães, Portugal; 4Hospital dos SAMS de Lisboa, 1849-017 Lisboa, Portugal; ana.cat.angelo@gmail.com

**Keywords:** rotator cuff tear, arthroscopic superior capsular reconstruction, fascia lata graft, shoulder stability, musculoskeletal model

## Abstract

**Simple Summary:**

In arthroscopic superior capsular reconstruction (ASCR) in irreparable rotator cuff tears (IRCTs), a graft is positioned and fixed between the superior rim of the glenoid and the humeral supraspinatus footprint. The fixation of the graft aims to restore the stability and improve the kinematics of the shoulder. The shoulder position during fixation of the graft may be a key factor impacting the outcome of ASCR; however, biomechanical evidence is lacking, as most studies addressing ASCR have been conducted in cadavers. In this study, graft strain and glenohumeral joint reaction force, estimated using a 3-D musculoskeletal model of the upper limb, were used to evaluate graft integrity and shoulder stability, respectively. The results suggest that ASCR significantly improved shoulder stability compared to the preoperative condition; however, the shoulder positions of fixation associated with the greatest improvements were also associated with the highest risk of compromising the integrity of the graft due to high strains. This study provides new and important information regarding the role of shoulder positioning during fixation of the graft.

**Abstract:**

The shoulder position during fixation of the graft may be a key factor impacting the outcome of arthroscopic superior capsular reconstruction (ASCR) in irreparable rotator cuff tears (IRCTs). However, biomechanical evidence regarding this effect is lacking. The aim of this study was to evaluate the influence of the shoulder position during fixation of the graft on shoulder stability and graft tear risk in ASCR. A 3-D musculoskeletal model of the upper limb was modified to account for the fixation of the graft in ASCR, assuming a full-thickness tear of the supraspinatus tendon. The concomitant tenotomy of the long head of the biceps (LHB) tendon was also studied. The biomechanical parameters evaluated included the strain of the graft and the glenohumeral joint reaction force (GH JRF), which were used to evaluate graft integrity and shoulder stability, respectively. Fixation of the graft considering abduction angles greater than 15° resulted in a high risk for graft tearing when the arm was adducted to the side of the trunk. For abduction angles below 15°, the mean shoulder stability improved significantly, ranging between 6% and 20% (*p* < 0.001), compared with that in the preoperative condition. The concomitant tenotomy of the LHB tendon resulted in loss of stability when compared to ASCR with an intact LHB tendon. The position of the shoulder during fixation of the graft has a significant effect on shoulder stability and graft tear risk after ASCR in IRCTs. This study provides new and important information regarding the role of shoulder positioning during fixation of the graft.

## 1. Introduction

The shoulder complex represents a perfect balance between mobility and stability [1,2,3,4], which is mainly provided by the combined action of the rotator cuff muscles and the superior capsule [3,5,6,7,8,9,10]. In elderly and low-demand patients, the preferred treatment option for intractable shoulder pain and irreparable RCTs (IRCTs) is reverse total shoulder arthroplasty, which meets patients’ functional demands and effectively reduces shoulder pain [11,12,13,14]. However, for the treatment of younger, high-demand patients, there is no gold standard, and most treatment options have pitfalls [11,12,13,14]. Recently, arthroscopic superior capsular reconstruction (ASCR) was introduced by Mihata et al. [15] as an alternative treatment option for IRCTs. A graft is positioned and fixed between the superior rim of the glenoid and the humeral supraspinatus footprint, with the aim of restoring the stability and physiological kinematics of the shoulder [16,17]. During this procedure, a concomitant tenotomy or tenodesis of the long head of the biceps (LHB) tendon may be performed, which has been shown to contribute to the relief of shoulder pain [18,19,20]. ASCR has been shown to produce excellent clinical outcomes [21]; however, graft tear rates can range from 4.2% to 75% [16,17,22,23,24]. ASCR using a human dermal allograft presents higher tear rates than ASCR using a fascia lata autograft [23,24]. Graft tears of larger dimensions have been correlated with decreased functional outcomes compared to functionality in the intact graft condition [25].

Further research is necessary to improve the medium- to long-term outcomes of ASCR [21]. The position of the shoulder for the fixation of the graft may be a key factor for the improvement of the outcomes of ASCR. However, its impact on the stability of the shoulder lacks evidence, and most studies addressing ASCR have been conducted in cadavers [26,27]. To the best knowledge of the authors of the current study, only a single computational study has reported the correlation between initial dermal allograft length, defined by the shoulder position of fixation, and graft strain [28]. No computational data exist regarding the effect of the position of the shoulder during the fixation of the graft on the stability of the joint. Moreover, the functional role of the graft remains unknown.

The purpose of this study was to evaluate the effect of the shoulder position during fixation of the fascia lata graft on graft tear risk and shoulder stability after ASCR for the treatment of irreparable full-thickness tears of the supraspinatus tendon. The hypotheses were that ASCR would increase the stability of the shoulder, and different positions of the shoulder during graft fixation would influence graft tear risk and shoulder stability.

## 2. Materials and Methods

### 2.1. Computational Modelling

The musculoskeletal model of the upper limb developed by Quental et al. [29,30] was used to investigate the effect of the fixation of the graft on shoulder biomechanics. The model is composed of 7 rigid bodies, including thorax, rib cage, clavicle, scapula, humerus, ulna, and radius, which are actuated by the main muscles of the upper limb, including the main dynamic stabilizers of the glenohumeral (GH) joint. The GH joint is modelled as a 3-degree-of-freedom spherical joint. For the introduction of the graft in the musculoskeletal model, a 3-D geometric model of the upper limb, based on the same subject of the musculoskeletal model, was modified according to the ASCR procedure using Solidworks (Dassault Systèmes, Waltham, MA, USA). With the guidance of two experienced shoulder orthopedic surgeons (C.d.C.A. and A.C.A.), the anchors’ site was chosen to be as similar as possible to that which is typically selected in the ASCR setting [14]. Two holes were modelled on the superior glenoid rim, as well as on the supraspinatus footprint. Considering the positioning of the holes, a single-piece graft was modelled as a set of four parallel segments fully connected to the bone at the origin and insertion sites, as shown in Figure 1. This segmentation allowed for the evaluation of the graft along its anterior–posterior direction [28]. Data for the four segments, including origin and insertion sites, were introduced into the musculoskeletal model for the simulation of the graft geometry. The path of graft segments was computed as a series of straight and curved lines that accounted for the possible wrapping of the graft around the humeral head, assumed as a sphere. The sum of the length of all these lines defined the length of each graft segment.

From the mechanical point of view, the graft was modelled as a group of passive elastic elements, whose forces depended exclusively on their length. The magnitude of the force exerted by each segment was obtained by multiplying their stress and cross-sectional area, defined as a quarter of the cross-sectional area of the single-piece graft, with 2.5 cm of width and 0.5 cm of thickness [19]. Graft material properties were defined based on experimental grip-to-grip tensile tests performed previously with samples of fascia lata, collected from the mid-thigh of fresh cadavers [31]. Each sample was folded three times, and a final six-layered graft was prepared, replicating the final graft used in the ASCR using the mid-thigh fascia lata autograft. To describe the non-linear relationship between the collected strain and stress data, a bilinear constitutive model was considered:(1)$$\left\{\begin{array}{l} \sigma =0\\ {\sigma =E}_{0}\varepsilon \\ \sigma =E\left(\varepsilon -\varepsilon^{*}\right){+E}_{0}\varepsilon^{*} \end{array}\right.\;\mathrm{for}\;\begin{array}{c} \varepsilon \;<\;0\\ \;0\leq \varepsilon \;\leq {\;\varepsilon }^{*}\\ \varepsilon \;>\;\varepsilon^{*} \end{array},$$
where σ is stress, ε is strain, E_0_ and E are, respectively, the modulus of elasticity of the toe and linear regions, and ε* is the transition strain. The constitutive parameters ε*, E_0_, and E, defined as 6.63%, 0.83 MPa, and 44.03 MPa, respectively, were computed as the average constitutive parameters estimated for all samples by fitting the constitutive model to the experimental data. The strain of each segment was defined as the change in graft length divided by its initial length. For each segment, the initial length was assumed as the graft path length upon fixation [28].

### 2.2. Shoulder Positions

To evaluate the impact of the shoulder position during the fixation of the graft on the shoulder biomechanics, different shoulder positions of fixation were simulated. The domain of the parameters to be tested was defined based on the typical arthroscopic settings of ASCR [8]. Abduction, relative to the trunk, was set between 0° and 30°, with 5° intervals. Forward flexion included each 15° position between 0° to 60° and the 70° position, which was based on the study conducted by de Campos Azevedo et al. [19]. For each combination of abduction and forward flexion, neutral rotation and 10° of internal and external rotation were tested. A total of 126 shoulder positions for the fixation of the graft were obtained. For the sake of simplicity, these positions are hereafter described as “(abduction, forward flexion, axial rotation)”, in degrees, where positive axial rotations describe internal rotations. For each position of the shoulder for the fixation of the graft, the orientation of the shoulder girdle was estimated using the regression equations of the scapular–humeral rhythm proposed by Xu et al. [32].

Considering that the position of the shoulder for the fixation of the graft should not be associated with a high risk of graft tear when the arm reaches its rest position of neutral abduction and forward flexion [33], a screening of the 126 proposed shoulder positions of fixation was performed to identify those for which at least one segment of the graft construct presented strains above the 15% failure limit [34,35]. Shoulder stability was not evaluated for these positions.

### 2.3. Musculoskeletal Simulations

To evaluate the relationship of the shoulder position for fixation of the graft and shoulder stability, inverse dynamic analyses were performed to estimate muscle and joint reaction forces. The redundant muscle force sharing problem was solved as the minimization of the muscle energy consumption subjected to the physiological boundaries of the muscle forces, the fulfilment of the equations of motion, and the stability of the GH and scapulothoracic joints [30,36]. The stability of the GH joint was ensured by limiting the ratios between the shear and the compressive components of the GH joint reaction (JRF) force along the directions depicted in Figure 2. The stability ratios (SRs) define the maximum allowed ratios before dislocation of the joint occurs. When the stability constraint of the GH joint is compromised, an additional compressive force is necessary to ensure that the ratio between the shear and the compressive components of the GH JRF vector lie within the allowed SR. The additional compressive force is achieved by increasing muscle activity. The SRs were taken from an intact labrum condition [37], since ASCR does not usually involve labrum excision. For the simulations of inverse dynamics, kinematic data of the upper limb were collected from the database of the Laboratory of Biomechanics of Lisbon (LBL) for 18 healthy subjects (7 males, 11 females; mean age, 24.5 ± 8.4 years; range, 18–55 years). The kinematic data were acquired using a Motion Capture System, from Qualisys (Gothenburg, Sweden) (ProReflex 500/1000), at a sampling frequency of 100 Hz. An acromion marker cluster was used to track the dynamic motion of the scapula [30]. These data included three motion trials of abduction in the frontal plane, forward flexion in the sagittal plane, and two activities of daily life (ADLs), namely, reaching the back and combing the hair. Overall, a total of 216 trials were included in this study.

For each of the 216 trials selected, 5 shoulder conditions were evaluated: (1) a healthy condition, with no muscle pathology; (2) a preoperative condition, with a full-thickness supraspinatus tendon tear; (3) a preoperative condition, with a full-thickness supraspinatus tendon tear and with an LHB tendon that no longer transmitted force; (4) and (5) postoperative conditions in which the graft was implanted to treat conditions (2) and (3), respectively. The postoperative condition (5) simulated ASCR with a concomitant tenotomy of the LHB tendon. The inability of the tendons to transmit force was simulated by removing the respective muscles from the musculoskeletal model. For each condition and trial, the graft strain, muscle forces, and JRFs at the GH joint were estimated for each motion instant. An estimate of shoulder stability, hereafter referred to as stability index (SInd), was computed as follows:(2)$$\mathrm{SIn}d=\frac{\frac{JRF_{shear}^{GH}}{JRF_{comp}^{GH}}}{\mathrm{SR}}\times \left(\lambda_{GH}+1\right),$$
where $JRF_{shear}^{GH}$ and $JRF_{comp}^{GH}$ correspond to the magnitude of the shear and the compressive components, respectively, of the GH JRF. The ratio between these components was normalized by the maximum stability threshold allowed for each direction on the glenoid plane, defined in detail in Figure 2, and the highest normalized ratio among all directions was selected for the SInd. The variable *λ_GH_* represents the Lagrange multiplier associated with the stability constraint; in this study, it was an output of the optimization algorithm used, the *fmincon* function of Matlab 2018a (MathWorks, Natick, MA, USA). When *λ_GH_* is equal to 0, the GH stability condition is naturally satisfied. A *λ_GH_* different than zero means that an additional compressive force is necessary to maintain the stability of the shoulder joint. When SInd is between 0 and 1, the non-dislocation requirements of the joint are naturally respected. When it is higher than 1, SInd provides an estimate of how far the stability would be from the maximum stability threshold if no stability constraint exists [38].

### 2.4. Biomechanical Evaluation of ASCR

Each motion trial was evaluated until the last motion instant where the stress of at least one of the shoulder positions for the fixation of the graft was not null. All other motion instants were disregarded because the graft worked only as a spacer, regardless of the shoulder position of fixation. When the strain of at least one segment exceeded the strain failure of 15%, the motion instants were assumed as “high-risk-of-tear-motion instants” and were considered a worst-case scenario in which the graft did not work due to failure. From a quantitative point of view, strains of 15% were assigned to all graft segments, and the SInds and GH JRFs from the corresponding no-graft conditions were assigned to these instants of the motion. The incidence of the high-risk-of-tear-motion instants was evaluated between movements and segments of the graft.

To compare the biomechanical performance between the different shoulder conditions and shoulder positions of fixation, three criteria, based on strain, stability, and force, were implemented. The strain criterion provided the mean graft strain for all evaluated motion instants. The risk of failure increases as the strain criterion approaches the failure strain of 15% [34,35]. The stability and force criteria were used to estimate if ASCR could provide a condition close to the healthy condition, regarding stability and the GH JRF. These criteria provided the relative difference between the postoperative and the healthy conditions for the SInd and GH JFR. In the computational model used, stability and force criteria values close to 0% represent a condition in which the biomechanics of the shoulder is similar to that in healthy condition.

### 2.5. Statistical Analysis

One-way analysis of variance (ANOVA) and Tukey’s multiple comparison test [39] were used to compare the strain and stability criteria among shoulder positions of fixation. The tests were conducted in Matlab 2018a using the *anova1* and *multcompare* functions. Each analysis was performed with the significance level set to *p* = 0.05.

## 3. Results

### 3.1. Risk of Tear

Of the 126 proposed shoulder positions of fixation, 81 failed the screening step. As shown in Figure 3, high degrees of shoulder abduction during fixation of the graft failed to reach the rest position without exceeding the failure strain. For the 45 shoulder positions of fixation that passed the screening step, almost 36% of the evaluated motion instants presented a high risk of tear. Of these, 72%, 16%, 6%, and 6% belonged to the movements reaching the back, abduction, forward flexion, and combing the hair, respectively. Considering the graft segments individually, the highest incidence of a high risk of tear was observed for the most posterior segment, for which strains exceeded the failure strain in almost 30% of the evaluated motion instants. On average, when at the risk of failure, half of the graft (two adjacent segments) were at risk, regardless of the shoulder positions of fixation. These results are further detailed in the Appendix A.

### 3.2. Biomechanical Performance of ASCR

The strain, stability, and force criteria are presented in Figure 4 in the form of heatmaps for the shoulder condition (4), with the fascia lata graft implanted and a full-thickness tear of the supraspinatus. As no relevant qualitative differences were found for the shoulder condition (5), for the sake of briefness, its results are only presented in the Appendix A. The shoulder positions for the fixation of the graft with 5° or 10° of abduction and 10° of internal rotation were associated with the greatest improvements in shoulder stability; however, these were also the positions of fixation associated with the highest strains and GH JRFs. For both postoperative conditions (4) and (5), the mean graft strains ranged between 3% and 8%. The GH JRFs increased between 1% and 10% compared to those in healthy condition. Shoulder positions of fixation with internal rotation were more stable than the respective positions of fixation with neutral and external rotation of the humerus.

The results of the multiple comparison analysis of the strain and stability criteria for the shoulder conditions (4) and (5) are presented in Figure 5. Relative to the healthy shoulder condition, shoulder stability was estimated to decrease by 17% and 37% after a full-thickness tear of the supraspinatus tendon and full-thickness tears of both supraspinatus and LHB tendons, respectively. Compared to the respective preoperative conditions, a significant improvement (*p* < 0.05) in shoulder stability was found after ASCR. This improvement varied between 6% and 15% for the shoulder condition (4) and between 9% and 20% for the shoulder condition (5). Regardless of the shoulder position for the fixation of the graft, the presence of the LHB was always associated with higher stability compared to the condition in which the LHB was absent.

## 4. Discussion

ASCR is a novel and not yet fully understood procedure, and the position of the shoulder for fixation of the graft may be a key factor impacting the outcome of the procedure. The main findings of this study were that initial graft positioning influenced graft tear risk and shoulder stability. Moreover, regardless of the shoulder position of fixation, ASCR improved shoulder stability, confirming the hypotheses of this study. The role of the graft as a stabilizer of the shoulder, working more than a spacer against superior translations, was demonstrated.

The full-thickness tear of the supraspinatus tendon resulted in a loss of shoulder stability, compared to the healthy condition. In the study of Steenbrink et al. [40] in which the influence of RCTs on shoulder stability was investigated, an isolated tear of the supraspinatus did not cause significant differences from the healthy condition. However, the stability criterion used by Steenbrink et al. [40] was less restrictive than that used in the current study. Moreover, their findings were based on a single shoulder position.

Shoulder positions during fixation ranging from 20° to 30° of abduction were estimated to represent a high risk of graft tear when the arm reaches the resting position. These results are in accordance with previous cadaveric studies which recommended against attaching the graft at high degrees of abduction [26,27], even though their upper bound (of 45° of abduction) was higher. A conservative evaluation of the risk of graft tear was chosen in the screening step, assuming that no graft segment could present strains above the failure strain because graft tears are a complication of ASCR which surgeons aim to avoid [24]. In the cadaveric studies conducted by Mihata et al. [26] and Adams et al. [27], the graft was evaluated as a single piece, and the possibility of partial or micro-tears was not considered. The musculoskeletal model applied in the current study allowed the authors to assess four portions of the graft individually, anticipating the existence of partial graft tears and allowing the analysis of their effect on the long-term integrity of the graft as a whole. Motions involving excessive axial rotations for low forward flexion angles and for extensions of the arm behind the back were also found to be associated with a high risk of graft tear, which is consistent with the findings of Hast et al. [28]. Rehabilitation after ASCR needs to be carefully designed and should consider these data to avoid compromising the integrity of the graft.

The results demonstrated the role of the graft as a stabilizer of the shoulder. When the graft deforms, it produces forces that assist the stabilization of the joint by compressing and centering the humeral head in the glenoid cavity [14]. The contribution of the graft to shoulder stability occurred mainly in low ranges of motion of abduction and forward flexion and in the initial and final phases of the reaching the back motion, during which the axial rotation of the arm and the combined extension and axial rotation of the arm, respectively, deformed the graft. This stabilization of the shoulder came at the cost of larger graft strains and GH JRFs. Although the computational model used in the current study could not estimate superior translations, the role of the graft as a spacer has been reported in previous cadaveric studies [26,27,41].

Shoulder positions of fixation that better improved the stability of the shoulder joint were associated with high strains and with a large number of high-risk-of-tear-motion instants. When at risk, on average, more than half of the graft presented strains above the failure limit. Fixation of the graft with the shoulder at 5° to 10° of abduction and 10° of internal rotation correlated with the most significantly improved shoulder stability compared to the preoperative condition (*p* < 0.001). To avoid graft tearing, a shoulder position of fixation that presented a low incidence of high-risk-of-tear-motion instants, but for which the stability improvements were significant compared to the healthy condition, should be selected.

Another finding of this study was the influence of the concomitant tenotomy of the LHB tendon on shoulder stability. The application of both ASCR and concomitant tenotomy of the LHB tendon to a shoulder condition, where the supraspinatus was completely torn and the LHB tendon was still able to transmit force, was not able to improve the stability of the shoulder joint relative to the preoperative condition. Only for a small group of shoulder positions of fixation did shoulder stability not differ from that in the preoperative condition. This finding supports the importance of the LHB tendon as a stabilizer of the GH joint and that the isolated tenotomy of the LHB tendon may lead to worst functional outcomes [42,43,44,45]. From a qualitative point of view, for the remaining shoulder positions of fixation, for which the tenotomy of the LHB tendon led to a significant loss of shoulder stability (*p* = 0.007), similar findings were obtained for the shoulder conditions (4) and (5). The wear of the LHB tendon is mainly associated with the loss of the stabilizing function of the supraspinatus [20,46,47,48], which means that the LHB is likely to be confirmed to be frail or fully torn during ASCR. Furthermore, the tenotomy of the LHB tendon in ASCR may be associated with a significant improvement of shoulder pain [19]. Accordingly, despite the musculoskeletal model predicting a loss of shoulder stability, the clinical improvements that are not accounted for by the model may justify the concomitant tenotomy of the LHB tendon in ASCR.

The current study provides insight into the restoration of the shoulder biomechanics after ASCR, and studies addressing potential technical improvements may translate into better clinical outcomes. Nevertheless, the results should be interpreted considering the limitations of this study. The failure strain considered for the fascia lata graft construct was based on data for single-layer fascia lata specimens [34,35]. Further experimental studies are necessary to assess if the single-layer fascia lata data regarding failure strain apply to the fascia lata graft constructs. Healing of the graft was not modelled, and the graft was assumed to be perfectly bonded to the bone, as in a previous computational study [28].

The GH joint was modelled as an ideal spherical joint, i.e., without allowing translations of the humeral head, and the computational model used in the current study could not estimate superior translations [26,27]. Nevertheless, the methodology of the current study was considered adequate because the stability constraint applied during inverse dynamics is expected to simulate the stability requirements of the joint [49]. Furthermore, by modelling the shoulder joint as a spherical joint with clearance [50], the musculoskeletal model applied may be used in future studies to evaluate the role of the graft as a spacer.

The kinematic data applied were obtained from a young and healthy population, similarly to previous studies [28]. It remains unknown whether patients who undergo ASCR recover healthy shoulder kinematics after the rehabilitation period. Despite using kinematic data of different subjects, the anthropometric parameters of the musculoskeletal model were not scaled. This study aimed to directly compare different shoulder positions of fixation during ASCR, and scaling of the musculoskeletal model was not performed to limit changes to the positioning of the arm during graft implantation.

Finally, shoulder stability was only evaluated for full-thickness tears of the supraspinatus; therefore, further investigation is necessary regarding IRCTs extending to the remaining rotator cuff tendons.

## 5. Conclusions

ASCR significantly improved shoulder stability compared to the preoperative condition; however, the shoulder positions of fixation associated with the highest improvements were also associated with the highest risk of compromising the integrity of the graft due to high strains. The concomitant tenotomy of the LHB tendon was associated with a significant loss of stability when compared to ASCR with an intact LHB tendon.

## Figures and Tables

**Figure 1 biology-10-01263-f001:**
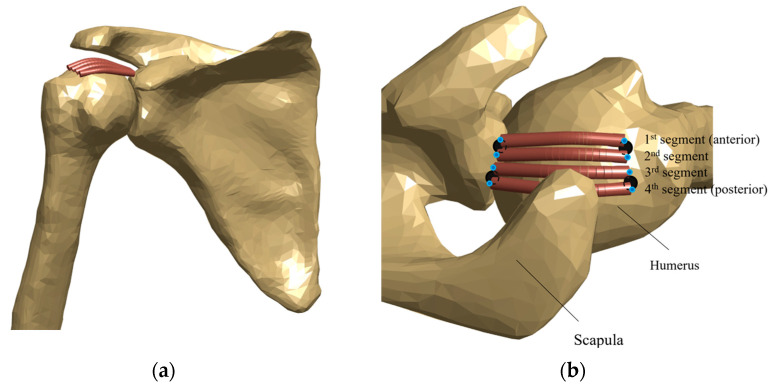
Graft path for each segment of the single-piece graft in (**a**) anterior and (**b**) superior views, for a specific position of the shoulder. In (**b**), black circles denote the anchor’s site, and blue circles denote the origin and insertion sites for each segment.

**Figure 2 biology-10-01263-f002:**
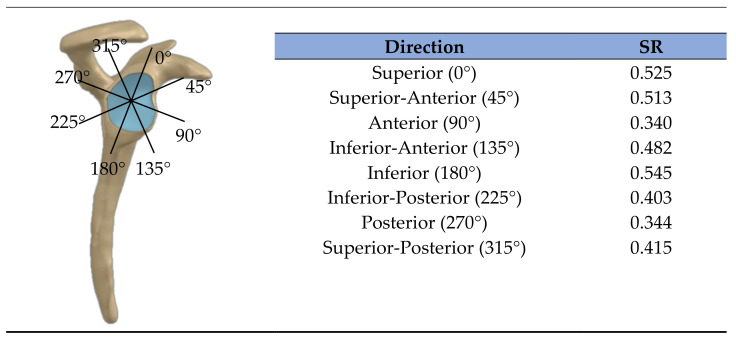
Stability ratios (SR) for eight directions of dislocation, in the glenoid plane, defined by the highest tolerable ratio between the shear and the compressive components of the GH reaction force.

**Figure 3 biology-10-01263-f003:**
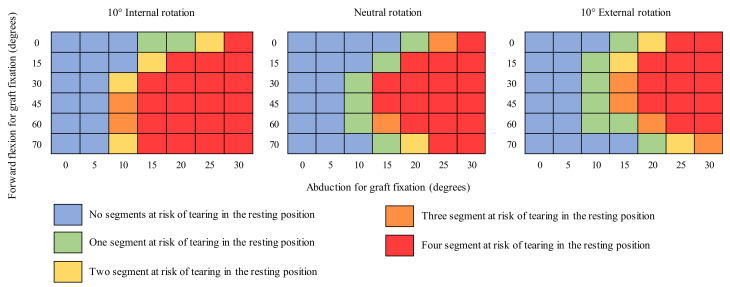
Shoulder positions of fixation that passed (blue) and failed the screening step, depending on the number of segments of the graft at risk of failure, one (green), two (yellow), three (orange), and four (red).

**Figure 4 biology-10-01263-f004:**
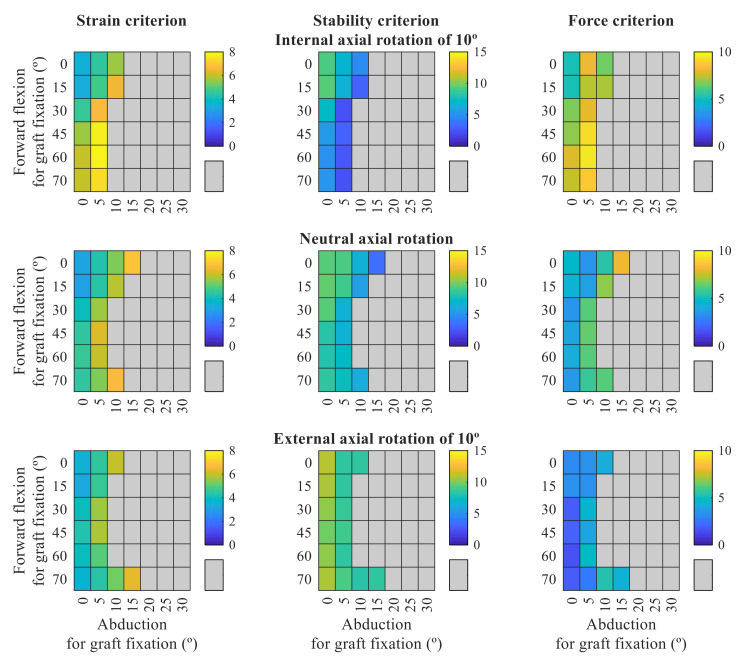
Heatmaps of the strain, stability, and force criteria for the shoulder condition (4), with implanted graft and a full-thickness tear of the supraspinatus. The risk of failure increases as the strain criterion approaches 15%. Stability and force criteria values closer to 0% mean that the graft can better restore the stability of the shoulder joint to its healthy condition. The grey color represents the shoulder positions of fixation that failed the screening step.

**Figure 5 biology-10-01263-f005:**
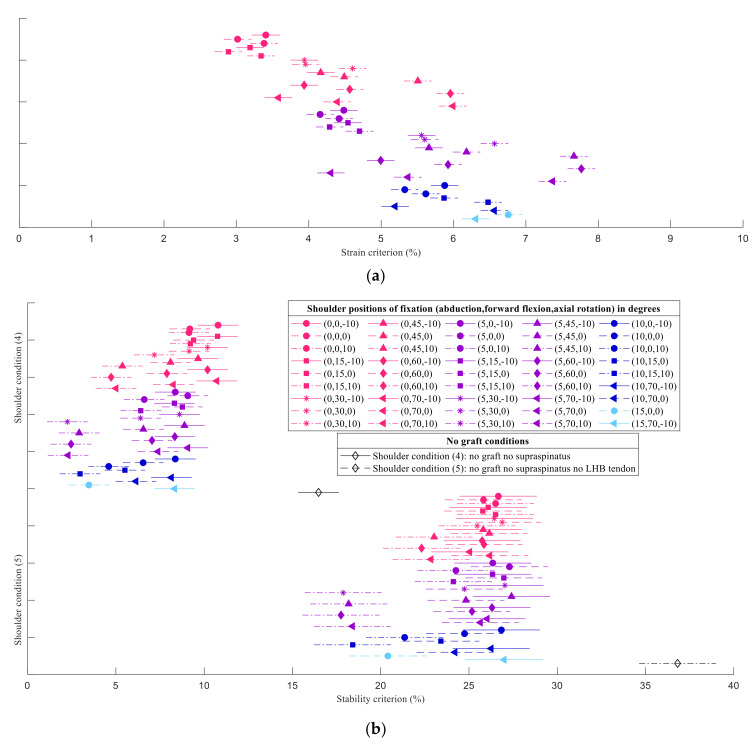
Multiple comparison test for the (**a**) strain criterion, equal for both shoulder conditions, and (**b**) stability criterion for the postoperative conditions, with a full-thickness tear of the supraspinatus tendon, condition (4), and with a full-thickness tear of the supraspinatus tendon and an LHB tendon that no longer transmitted force, condition (5). The markers represent the mean value of each group. The lines crossing the markers symbolize the respective comparison intervals. The difference between two means is statistically significant when the comparison intervals do not overlap. Shoulder positions of fixation are presented as “(abduction, forward flexion, axial rotation)”, in degrees, where positive axial rotations describe internal rotations. In (**b**), each shoulder condition is followed by the respective shoulder condition with no graft implanted. The healthy condition is represented by the stability criterion equal to 0%.

## Data Availability

Not applicable.

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
