# Peer review of "Shoulder Positioning during Superior Capsular Reconstruction: Computational Analysis of Graft Integrity and Shoulder Stability"

_biology, 2021, doi:10.3390/biology10121263_

Round 1

Reviewer 1 Report

In this paper authors investigate the problem of optimizing the positioning of the shoulder during superior capsular reconstruction. This is a very difficult task from a computational point of view. However, in my opinion, on the whole, authors have solved it. Thus, this article can be accepted for publication after some refinements.

  1. Authors use the musculoskeletal model developed by them earlier the study of which involves the optimization of some criterion. In this article and in their other articles, authors use the muscle energy consumption as such criterion. Indeed, this is a fairly plausible criterion, but why namely it was chosen to solve the problem of muscle redundancy? Why, for example, the kinematic criterion of least discomfort (K. Abdel-Malek), dynamic criterion of minimum torque-change (Y. Uno, et al.) or some other criterions were not taken into account?
  2. For the solving of the optimization problem, it becomes necessary to solve the inverse problem of the upper limb dynamics. From the text follows that the optimization method is given in the article [30], which contains criterion (6) containing indefinite Lagrange multipliers. It is clear that in the process of the solution these multipliers will be found and they have a certain physical meaning and numerical value. But what is the meaning of the λ in the SInd criterion (2)? How this number determined? Can we call it the Lagrange multiplier?
  3. To solve the optimization problem, authors used the WMIDO procedure. The following questions arise:

- convergent solutions are obtained in all cases?

- what is the accuracy of obtained solutions?

- is it necessary to describe in detail the structural, kinematic and dynamic model of the shoulder, as the authors do, or can be a simpler model used for this task?

Author Response

Response to Reviewer 1 Comments

We want to express our appreciation to the Reviewer for the constructive evaluation of our work that allowed us to improve it. In what follows, we discuss the changes performed to our work. We hope that our revision clearly addresses the Reviewer’s recommendations.

Remark 1: Authors use the musculoskeletal model developed by them earlier the study of which involves the optimization of some criterion. In this article and in their other articles, authors use the muscle energy consumption as such criterion. Indeed, this is a fairly plausible criterion, but why namely it was chosen to solve the problem of muscle redundancy? Why, for example, the kinematic criterion of least discomfort (K. Abdel-Malek), dynamic criterion of minimum torque-change (Y. Uno, et al.) or some other criterions were not taken into account?

Reply: The muscle redundancy problem could have been solved with different criteria, as stated by the Reviewer. However, since we aimed to investigate the effect of shoulder position during fixation of the fascia lata graft on graft tear risk and shoulder stability after ASCR, the evaluation of different criteria was out of the scope of the study. Considering that several physiological criteria have been proposed in the literature, and that  specific strategy adopted by the central nervous system for muscle recruitment is still unknown, the muscle energy consumption criterion was selected as the optimization criterion for its better consistency between experimental and computational results, compared to the most  used criterion in inverse dynamics (the quadratic stress cost function) [R1].

Remark 2: For the solving of the optimization problem, it becomes necessary to solve the inverse problem of the upper limb dynamics. From the text follows that the optimization method is given in the article [30], which contains criterion (6) containing indefinite Lagrange multipliers. It is clear that in the process of the solution these multipliers will be found and they have a certain physical meaning and numerical value. But what is the meaning of the λ in the SInd criterion (2)? How this number determined? Can we call it the Lagrange multiplier?

Reply: Thank you for your question as it allowed us to identify a possible source of misunderstanding in the methodology. The optimization problem must fulfill the equations of motion while searching for the optimum solution of the muscle redundancy problem. Among the unknown variables, which need to be determined, are the Lagrange multipliers associated with the kinematic constraints of the system (for instance, the spherical joint simulating the glenohumeral joint), which determine the internal forces of the musculoskeletal system. These Lagrange multipliers are described in detail, for instance, in [R2], but are omitted here for the sake of simplicity. The parameter  in the SInd criterion is also a Lagrange multiplier, but this Lagrange multiplier is related with the constraint of the optimization problem that imposes the stability requirements for the glenohumeral joint. Its value is determined during the optimization problem and is output along with the optimal solution. When λ is equal to 0, the GH stability condition is naturally satisfied, i.e., no additional force is required to compress the humeral head against the glenoid, while a λ different than zero means that an additional compressive force is necessary to maintain the stability of the shoulder joint. To make this clearer in the revised manuscript,  was changed to  in Equation 2, and the last paragraph of Section 2.3 was revised to:

“where  and  correspond to the magnitude of the shear and compressive components, respectively, of the GH JRF. The ratio between these components was normalized by the maximum stability threshold allowed for each direction on the glenoid plane, defined in detail in Figure. 2, and the highest normalized ratio among all directions was selected for the SInd. The variable λGH represents the Lagrange multiplier associated with the stability constraint; in this study, it was an output of the optimization algorithm used, the fmincon function of Matlab 2018a (MathWorks, Natick, MA, USA). When λGH is equal to 0, the GH stability condition is naturally satisfied. A λGH different than zero means that an addition-al compressive force was necessary to maintain the stability of the shoulder joint.” (Lines 179-188 of the revised manuscript)

Remark 3: To solve the optimization problem, authors used the WMIDO procedure. The following questions arise: convergent solutions are obtained in all cases? What is the accuracy of obtained solutions? Is it necessary to describe in detail the structural kinematic and dynamic model of the shoulder, as the authors do, or can be a simpler model used for this task?

Reply: The optimization problem was solved using static optimization and all evaluated motion instants (for all subjects and trials) converged. Considering that in vivo measurements of muscle and joint reaction forces are limited, verifying the accuracy of computational musculoskeletal models is extremely difficult [R3]. Yet, the musculoskeletal model of the upper limb applied in this study was qualitatively validated by comparing muscle force predictions with EMG data and glenohumeral joint reaction forces with in vivo data measured using instrumented prostheses [R2,R4], which provides confidence in its application.

Regarding the level of complexity of the musculoskeletal model considered, we believe it to be necessary to simulate, as best as possible, the complex biomechanics of the shoulder [R5]. Similar levels of complexity are considered by other detailed musculoskeletal models of the upper limb [R6,R7]. Simplifications like modelling parts of the shoulder mechanism or neglecting the scapulothoracic joint are likely to impose serious limitations in the simulation of the shoulder biomechanics [R3].

References

[R1]   Nikooyan AA, Veeger H, Westerhoff P, Bergmann G, Van Der Helm FCT. Relative contribution of different muscle energy consumption processes in an energy-based muscle load sharing cost function. J Mech Med Biol 2013;13:1–18. https://doi.org/10.1142/S0219519413500097.

[R2]   Quental C, Azevedo M, Ambrósio J, Goncalves SB, Folgado J. Influence of the Musculotendon Dynamics on the Muscle Force-Sharing Problem of the Shoulder-A Fully Inverse Dynamics Approach. J Biomech Eng 2018;140. https://doi.org/10.1115/1.4039675.

[R3]   Bolsterlee B, Veeger DHEJ, Chadwick EK. Clinical applications of musculoskeletal modelling for the shoulder and upper limb. Med Biol Eng Comput 2013;51:953–63. https://doi.org/10.1007/s11517-013-1099-5.

[R4]   Quental C, Folgado J, Ambrósio J, Monteiro J. Critical analysis of musculoskeletal modelling complexity in multibody biomechanical models of the upper limb. Comput Methods Biomech Biomed Engin 2015;18. https://doi.org/10.1080/10255842.2013.845879.

[R5]   Veeger HEJ, van der Helm FCT. Shoulder function: The perfect compromise between mobility and stability. J Biomech 2007;40:2119–29. https://doi.org/10.1016/j.jbiomech.2006.10.016.

[R6]   Charlton IW, Johnson GR. A model for the prediction of the forces at the glenohumeral joint. Proc Inst Mech Eng Part H J Eng Med 2006;220:801–12. https://doi.org/10.1243/09544119JEIM147.

[R7]   Asadi Nikooyan A, Veeger HEJ, Chadwick EKJ, Praagman M, Van Der Helm FCT. Development of a comprehensive musculoskeletal model of the shoulder and elbow. Med Biol Eng Comput 2011;49:1425–35. https://doi.org/10.1007/s11517-011-0839-7.

Reviewer 2 Report

The manuscript “Shoulder positioning during superior capsular reconstruction: computational analysis of graft integrity and shoulder stability” by Madalena Antunes, Carlos Quental , João Folgado, Clara de Campos Azevedo  and Ana Ângelo  is an article that aimed to evaluate the effect of shoulder position during fixation of the fascia lata graft on graft tear risk and shoulder stability after ASCR for the treatment of irreparable full-thickness tears of the supraspinatus tendon

Below are my comments and remarks regarding the article:

1. Perfect sphericity may limit appropriate measurements
2. Anthropometric parameters and other features, e.g. the narrowing of the subacromial space has been determined, may influence the ACSR
3. What kind of fixation was used?

Author Response

Response to Reviewer 2 Comments

We want to express our appreciation to the Reviewer for the constructive evaluation of our work. In what follows, we address the Reviewer’s concerns.

Remark 1: Perfect sphericity may limit appropriate measurements.

Reply: We agree with the Reviewer that the assumption of an ideal spherical joint for the glenohumeral joint constituted one of the limitations of this study, as it precluded the estimation of joint translations. Nonetheless, the methodology followed was considered adequate for the aim of this study due to the stability constraint considered for the glenohumeral joint, which is expected to simulate its stability requirements [R1]. In future studies, a more complex description of the shoulder joint may be used to evaluate the role of the graft as a spacer. This limitation is discussed in the manuscript in the antepenultimate paragraph of Section 4 (Discussion):

“The GH joint was modelled as an ideal spherical joint, i.e., without allowing translations of the humeral head, and the computational model used in the current study could not estimate superior translations [26,27]. Nevertheless, the methodology of the current study was considered adequate because the stability constraint applied during inverse dynamics is expected to simulate the stability requirements of the joint [49]. Furthermore, by modelling the shoulder joint as a spherical joint with clearance [50], the musculoskeletal model applied may be used in future studies to evaluate the role of the graft as a spacer.” (Lines 349-355 of the revised manuscript)

Remark 2: Anthropometric parameters and other features, e.g. the narrowing of the subacromial space has been determined, may influence the ACSR.

Reply: We agree that anthropometric parameters may influence the ASCR. However, because we aimed to directly compare different shoulder positions of fixation during ASCR, disregarding anthropometric variability allowed a direct evaluation of results, as the positioning of the shoulder during graft implantation was the only factor changing between simulations.

Remark 3: What kind of fixation was used?

Reply: We modeled the ASCR assuming a perfect bonding between the graft and bone, similar to a previous computational study of ASCR [R2]. The modeling of the graft, including its connection to the bone, is described in the first paragraph of Section 2.1:

“With the guidance of two experienced shoulder orthopedic surgeons (C.d.C.A. and A.A.), the anchors’ site was chosen to be as similar as possible to that which is typically selected in the ASCR setting [14]. Two holes were modelled on the superior glenoid rim, as well as on the supraspinatus footprint. Considering the positioning of the holes, a single-piece graft was modelled as a set of four parallel segments fully connected to the bone at the origin and insertion sites, as shown in Figure. 1. This segmentation allowed for the evaluation of the graft along its anterior-posterior direction [28].” (Lines 89-96 of the revised manuscript)

References

[R1]   Dickerson CR, Chaffin DB, Hughes RE. A mathematical musculoskeletal shoulder model for proactive ergonomic analysis. Comput Methods Biomech Biomed Engin 2007;10:389–400. https://doi.org/10.1080/10255840701592727.

[R2]   Hast MW, Schmidt EC, Kelly JD, Baxter JR. Computational optimization of graft tension in simulated superior capsule reconstructions. J Orthop Res 2018;36:2789–96. https://doi.org/10.1002/jor.24050.

Reviewer 3 Report

Lines 15/16, I would suggest changing "restore (...) physiological kinematics" to "to restore the stability and improve kinematics". 

Line 19, adding "3-D" before "musculoskeletal model" would clarify method used. Same in the Line 29.

Line 275. Please extend the sentence "The role of the graft as more than a spacer working against superior translations was demonstrated. " I do agree with it but it woud be interesting to discuss this topic. Do you mean stabilizer function as discussed below or other functions, too?

Lines 331-332. I would suggest to discuss tenotomy and tenodesis in relation to recent papers on this topic (i.e. doi: 10.1016/j.jse.2021.02.002.; 10.1007/s00167-015-3640-6; 10.3390/jcm9123938.) I do agree that LHBT is important to the  shoulder function after SCRs but it is still unclear if it is crucial after all other surgieries due to RCTs. It should be clearly stated that the authors are writing about the role of LHBT after SCR.

Author Response

Response to Reviewer 3 Comments

We want to express our appreciation to the Reviewer for the constructive evaluation of our work that allowed us to improve it. In what follows, we discuss the changes performed to our work. We hope that our revision clearly addresses the Reviewer’s recommendations.

Remark 1: Lines 15/16, I would suggest changing "restore (...) physiological kinematics" to "to restore the stability and improve kinematics".

Reply: Thank you for your suggestion. As suggested, the referred lines were revised to:

“The fixation of the graft aims to restore the stability and improve kinematics of the shoulder.” (Lines 15-16 of the revised manuscript)

Remark 2: Line 19, adding "3-D" before "musculoskeletal model" would clarify method used. Same in the Line 29.

Reply: The manuscript was revised as suggested:

“In this study, graft strain and glenohumeral joint reaction force, estimated using a 3-D musculoskeletal model of the upper limb, were used to evaluate graft integrity and shoulder stability, respectively. The results suggest that ASCR significantly improved shoulder stability compared to the preoperative condition; however, the shoulder positions of fixation associated with the highest improvements were also associated with the highest risk of compromising the integrity of the graft due to high strains. This study provides new and important information regarding the role of shoulder position during fixation of the graft.

Abstract: The shoulder position during fixation of the graft may be a key factor impacting the outcome of arthroscopic superior capsular reconstruction (ASCR) in irreparable rotator cuff tears (IRCTs). However, biomechanical evidence regarding its effect is lacking. The aim of this study was to evaluate the influence of the shoulder position during fixation of the graft on shoulder stability and graft tear risk in ASCR. A 3-D musculoskeletal model of the upper limb was modified to account for the fixation of the graft in ASCR, assuming a full-thickness tear of the supraspinatus tendon.” (Lines 19-32 of the revised manuscript)

Remark 3: Line 275. Please extend the sentence "The role of the graft as more than a spacer working against superior translations was demonstrated. " I do agree with it but it would be interesting to discuss this topic. Do you mean stabilizer function as discussed below or other functions, too?

Reply: In the first paragraph of the discussion, we intended to summarize the main conclusions of this study, and thus we meant to refer the stabilizer function of the graft discussed in further detail later in the discussion. To clarify this, the first paragraph of Section 4 was revised to:

“ASCR is a novel and not yet fully understood procedure, and the position of the shoulder for fixation of the graft may be a key factor impacting the outcome of the procedure. The main findings of this study were that initial graft positioning influenced shoulder stability and graft tear risk. Moreover, regardless of the shoulder position of fixation, ASCR improved shoulder stability, confirming the hypotheses of this study. The role of the graft as a stabilizer of the shoulder, working more than a spacer against superior translations, was demonstrated.” (Lines 273-279 of the revised manuscript)

Remark 4: Lines 331-332. I would suggest to discuss tenotomy and tenodesis in relation to recent papers on this topic (i.e. doi: 10.1016/j.jse.2021.02.002.; 10.1007/s00167-015-3640-6; 10.3390/jcm9123938.) I do agree that LHBT is important to the shoulder function after SCRs but it is still unclear if it is crucial after all other surgeries due to RCTs. It should be clearly stated that the authors are writing about the role of LHBT after SCR.

Reply: Thank you for your remark. We agree that it should be clear that we are discussing the role of the LHB tendon after SCR. To address this, the sixth paragraph of Section 4 was modified to:

“Wear of the LHB tendon is mainly associated with the loss of the stabilizing function of the supraspinatus [20,46–48], which means that the LHB is likely to be confirmed to be frail or fully torn during the ASCR. Furthermore, the tenotomy of the LHB tendon in ASCR may be associated with a significant improvement of shoulder pain [19]. Accordingly, despite the musculoskeletal model predicting a loss of shoulder stability, the clinical improvements that are not accounted for by the model may justify the concomitant tenotomy of the LHB tendon in ASCR.” (Lines 333-339 of the revised manuscript)

For the sake of objectivity, we do not discuss differences between tenotomy and tenodesis as this subject is out of the scope of this study.

Reviewer 4 Report

This is a well conducted study and the manuscript is very well written

Author Response

Response to Reviewer 4 Comments

We want to express our appreciation to the Reviewer for the constructive evaluation of our work.

Round 2

Reviewer 2 Report

Thank you and I accept all explanations.